# Emerging spin–phonon coupling through cross-talk of two magnetic sublattices

Mads C. Weber [1,2,3,4✉], Mael Guennou [2,3], Donald M. Evans [5], Constance Toulouse [2,3], Arkadiy Simonov [1], Yevheniia Kholina [1], Xiaoxuan Ma [6], Wei Ren [6], Shixun Cao [6✉], Michael A. Carpenter[5], Brahim Dkhil [7], Manfred Fiebig [1] & Jens Kreisel [2,3]

Many material properties such as superconductivity, magnetoresistance or magnetoelectricity emerge from the non-linear interactions of spins and lattice/phonons. Hence, an in-depth understanding of spin–phonon coupling is at the heart of these properties. While most examples deal with one magnetic lattice only, the simultaneous presence of multiple magnetic orderings yield potentially unknown properties. We demonstrate a strong spin–phonon coupling in $SmFeO_3$ that emerges from the interaction of both, iron and samarium spins. We probe this coupling as a remarkably large shift of phonon frequencies and the appearance of new phonons. The spin–phonon coupling is absent for the magnetic ordering of iron alone but emerges with the additional ordering of the samarium spins. Intriguingly, this ordering is not spontaneous but induced by the iron magnetism. Our findings show an emergent phenomenon from the non-linear interaction by multiple orders, which do not need to occur spontaneously. This allows for a conceptually different approach in the search for yet unknown properties.

[1] Department of Materials, ETH Zurich, 8093 Zurich, Switzerland. [2] Department of Physics and Materials Science, University of Luxembourg, 41 Rue du Brill, L-4422 Belvaux, Luxembourg. [3] Materials Research and Technology Department, Luxembourg Institute of Science and Technology, 41 Rue du Brill, L-4422 Belvaux, Luxembourg. [4] Institut des Molécules et Matériaux du Mans, UMR 6283 CNRS, Le Mans Université, 72085 Le Mans, France. [5] Department of Earth Sciences, University of Cambridge, Downing Street, Cambridge CB2 3EQ, UK. [6] Department of Physics, Materials Genome Institute and International Center for Quantum and Molecular Structures, Shanghai University, Shanghai 200444, China. [7] Laboratoire Structures, Propriétés et Modélisation des Solides, Centrale Supélec, CNRS-UMR8580, Université Paris-Saclay, 91190 Gif-sur-Yvette, France. ✉email: mads.weber@univ-lemans.fr; sxcao@shu.edu.cn

The richness of physical phenomena in correlated oxides roots in the interaction and competition of coexisting properties and instabilities. A fundamental facet in materials with magnetic ions is the coupling of magnetic spins, the crystal lattice and lattice vibrations. From this interaction fascinating phenomena emerge, such as superconductivity[1], multiferroicity[2,3], giant thermal Hall effect[4] or ferroelectric phase transitions[5]. The presence of two magnetic ions in different sublattices makes the interaction particularly complex, but likewise intriguing. For instance, the cross-talk of transition-metal and rare-earth ions ($R^{3+}$) in complex oxides leads to phenomena such as spin-reorientations, magnetic compensation[6,7], solitonic lattices[8] or multiferroicity[9,10], including domain inversion[11] and interconversion of domains and domain walls[12,13]. At first sight, these phenomena seem to be of magnetic nature only. However, upon closer inspection, the coupling to the crystal lattice and related lattice vibrations turns out to be vital. For instance, multiferroicity entails an ionic displacement and tilts of the oxygen octahedra can give rise to a net-magnetization by a canting of the spins and thereby steer the rare-earth magnetism induced by the transition-metal ion[14]. Hence, the combination of the primary magnetic and structural orders is considerably more than the sum of its parts, manifesting in the emergence of enhanced or additional properties.

Despite the important interactions of magnetism and structure, spin–phonon coupling arising from the cross-talk of two magnetic ion subsystem remains largely unexplored. The observation and understanding of such cross-talk are at the heart of the present work. We show how the interaction of two magnetic sublattices in $SmFeO_3$ leads to the rise of an extraordinarily strong coupling between spins, lattice and lattice vibrations. First, for the high-temperature regime, we reveal a softening of the elastic moduli during the spin reorientation. Second, below room temperature, an unprecedentedly strong spin–phonon coupling arises from the non-spontaneous ordering of the $Sm^{3+}$ spins thanks to the exchange-field of the iron magnetism. Here we find strong indications that this "spin-spin–phonon" coupling gives rise to a phase change in $SmFeO_3$. Hence, the interaction of both magnetic sublattices drives a strongly non-linear material response—entirely absent for the individual magnetic sublattices.

## Results and discussion

$SmFeO_3$ crystallizes in a perovskite-type structure with the space group $Pnma$[7]. The primary structural distortions from the ideal perovskite structure are tilts of the oxygen octahedra, $a^-b^+a^-$ in Glazer's notation[15] (Fig. 1a). At $T_N = 680$ K, the iron spins order antiferromagnetically along the $c$-axis ($G_z$-type)[7]. A spin canting induces a weak-ferromagnetic moment along the $b$-axis ($F_y$) and an $A$-type component along the $a$-axis (expressed in Bertaut's notation[16]) (Fig. 1b). Between 450 K and 480 K, the iron-spin lattice experiences a spin reorientation to a $C_xG_yF_z$-type ordering triggered by the anisotropy change of the samarium moments[7]. Subsequently, the samarium spins align in the exchange field of iron. This becomes clear from magnetic measurements as a decline of the overall magnetization below 140 K[17,18] with a full compensation at 3.9 K[19]. Staub et al. showed that the $Fe^{3+}$ magnetism can even induce an antiferromagnetic order on the rare-earth sublattices below spin-reorientation[20]. Warshi et al. showed that this low-temperature ordering involves the formation of a cluster glass rather than a discrete transition[21]. The high temperatures of the magnetic ordering $T_N$ and the spin-reorientation[7] allow us to disentangle high- and low-temperature phenomena. This makes $SmFeO_3$ a model material, unlike other rare-earth transition-metal ions, in which most magnetic interactions occur far below room temperature.

To assess the interaction of spins and phonons, we performed Raman scattering complemented with resonant ultrasound spectroscopy (RUS) from 800 K down to 4 K. Both are excellent probes for detecting and tracing even subtle structural and magnetic changes[22,23]. Raman spectroscopy probes directly the optical phonons. Thanks to a recent work[24], we can assign all 24 Raman active phonons ($\Gamma = 7 A_g + 5 B_{1g} + 7 B_{2g} + 5 B_{3g}$[25]) to their specific vibrational patterns (see Fig. 2a and Supplementary Note 1). RUS provides a highly sensitive probe of static and dynamical lattice distortions that accompany relaxational processes and phase transitions[26,27]. In an RUS experiment, variations of elastic moduli scale with the square of the frequencies of individual resonances which are dominated by shearing motions, and acoustic loss is expressed in terms of the inverse mechanical quality factor, $Q^{-1}$. Combining both techniques provides access to the form and strength of both spin–phonon and spin-lattice coupling.

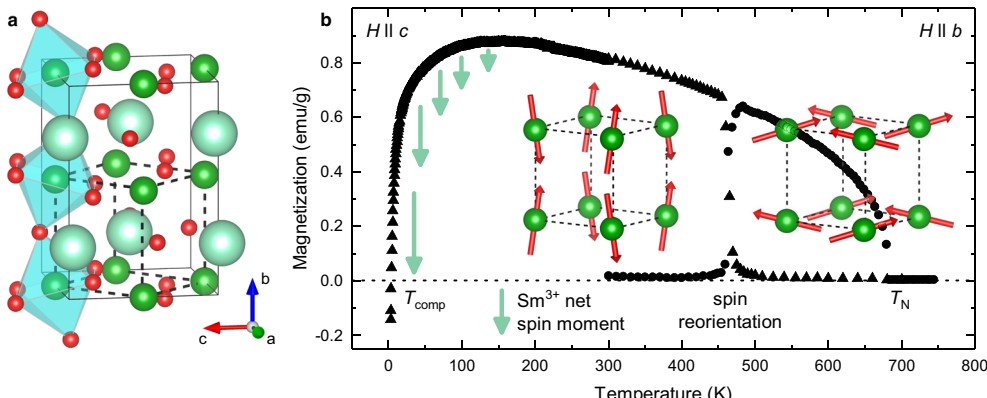

**Fig. 1 Magnetic properties of SmFeO₃. a** Rare-earth orthoferrite *Pnma* structure[55]. Oxygen, iron and rare-earth ions are given in red, green and turquoise, respectively, FeO₆ octahedra in pale blue. Solid and dashed lines describe the orthorhombic unit cell and the pseudo-cubic setting, respectively. **b** Evolution of the magnetization (data taken from Ref. [19]): at the Néel temperature $T_N$, the $Fe^{3+}$ spins order in a $A_xF_yG_z$-type fashion. At the spin reorientation, the magnetic order changes to $C_xG_yF_z$. The spin canting leads to a net-magnetization. Both $Fe^{3+}$ spin structures are sketched in pseudo-cubic settings. At low temperatures, the iron magnetism induces the magnetic $Sm^{3+}$ sublattice with a net-magnetic moment (turquoise arrows) aligning antiparallel to the iron moments, which leads to a magnetic compensation point $T_{comp}$ at 3.9 K.

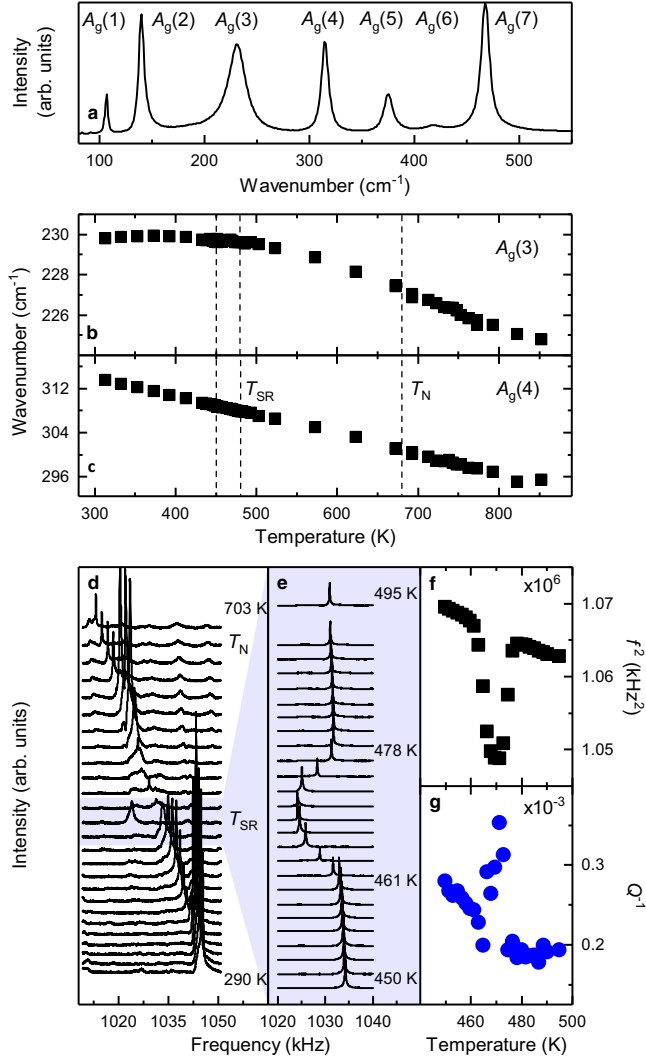

**Fig. 2 High-temperature evolution of lattice vibrations and elastic properties.**
**a** $A_g$-mode Raman spectrum of $SmFeO_3$ at room temperature in $z(yy)\bar{z}$ configuration using Porto's notation[37] (Symbols to the left and right of the parentheses denote the propagation direction, while symbols inside the parentheses indicate polarization of incident and scattered light, respectively.). **b**, **c** Evolution of the frequency of $A_g(3)$ and $A_g(4)$ Raman modes from 300 to 850 K, across the spin reorientation $T_{SR}$ and Néel $T_N$ temperatures. For the temperature evolution of the complete Raman spectra, the phonon frequencies and FWHM, please refer to Supplementary Note 1+2. **d** Temperature evolution of a representative resonance peak in RUS spectra from room temperature to 700 K, thus above the magnetic ordering (this behavior is alike for all ultrasound resonance bands; see Supplementary Figs. 5 and 6). **e** Magnification of the RUS spectra across the spin reorientation interval revealing a significant deviation in elastic moduli. **f** Evolution of the square of the frequency, $f^2$, which scales with some combination of single crystal elastic moduli, showing a softening through the reorientation transition. **g** The associated acoustic loss data, showing a peak in $Q^{-1}$ through the reorientation typical of mobile microstructure. For a further analysis of the elastic anomalies see Supplementary Note 4.

**Néel temperature ($T_N = 680$ K).** To understand the spin–phonon coupling, we start by investigating how the ordering of the iron spins at $T_N = 680$ K affects the vibrational system. When crossing $T_N$, both Raman and RUS data show no discernable discontinuity—or change in gradient—in either the wavenumber or the full width half maximum (FWHM) as well as the shear elastic moduli or the acoustic loss, respectively. This is

illustrated by two representative Raman bands (Fig. 2b, c and Supplementary Notes 1, 2) and an RUS acoustic resonance (Supplementary Fig. 4b). This indicates that spin-lattice coupling for the $Fe^{3+}$ magnetic order alone is very weak or absent, consistent with measurements of the lattice parameters[28]. The absence of spin–phonon coupling is surprising in the light of classical systems such as orthochromites and orthomanganites[29,30], where spin–phonon coupling occurs at $T_N$. This observation can be understood by the lack of Jahn-Teller distortion in orthoferrites, which, in these other cases, provides a stronger and direct link between the electronic subsystem and the crystal structure. We conclude that the magnetic order of iron alone has no detectable impact on the (an)elastic properties or the vibrational system.

**Spin reorientation ($T_{SR} = 450$ to 480 K).** With decreasing temperature, the magnetic $Sm^{3+}$ anisotropy changes. This change in anisotropy induces a rotation of the iron spin system[7] (Fig. 1b) marking the incipient cross-talk between the iron and the samarium magnetism. Unlike at $T_N$, the RUS data show elastic softening by up to a few percent and a closely correlated increase in acoustic loss between 460 and 480 K, before $f^2$ reverts to the same trend below 460 K as observed above 480 K (Fig. 2e–g, Supplementary Figs. 4–6). Observing the same temperature evolution of resonance frequencies before and after the onset of magnetic order means that the magnetic order parameter can only be very weakly coupled to strain – or that magnetoelastic coupling is completely absent, consistent with the literature and our findings at $T_N$[28,31]. Further, this lack of coupling of the magnetic order parameters with macroscopic strains together with the inverse correlations between the variations of $f^2$ and $Q^{-1}$, indicates that softening in the transition interval is due to anelastic relaxation.

The mechanism responsible for the concomitant changes of elastic moduli and acoustic losses is not known other than it involves anelastic relaxations of magnetoelastic origin. Strain relaxation of the structure in response to a dynamic stress is required, and one possibility is the existence of local regions with monoclinic distortions. However, although a monoclinic phase would occur in a continuous spin rotation[32], a coherent monoclinic distortion has not been detected by X-ray diffraction in orthoferrites[32,33]. Evidence from NMR spectroscopy that there could be local monoclinic symmetry in these two phases[34,35] has also been disputed[33]. (For further information see Supplementary Note 4).

In contrast to the strong RUS anomaly, frequencies of the optical Raman phonons show no observable change through the spin reorientation (Fig. 2b, c). This is consistent with the absence of changes in strain and the timescale of relaxational effects in the order of $\sim 10^{-6}$ s (ultrasound frequency), which would not be detected on the phonon timescale of $\sim 10^{-12}$ s. The FWHM (Supplementary Note 2) is characterized by a stagnation around the spin reorientation, and, in turn, the phonon lifetime does not lengthen any further. This indicates a reduced phonon correlation-length arising from a non-collective rotation of the iron spins and the competition of both magnetic phases (for further discussion see Supplementary Note 2). There is no evidence for a collective rotation and a resulting symmetry breaking on the phonon length scale, i.e., the intermediate length scale between the strictly local scale of NMR and the macroscopic length scale of X-ray diffraction.

To conclude the high temperature analysis, we find that the elastic moduli of $SmFeO_3$ soften significantly during the spin reorientation by a magnetoelastic relaxation mechanism present only in the reorientation state driven by the $Fe^{3+}$-$Sm^{3+}$-interplay. The magnitude of any macroscopic strains coupled with the

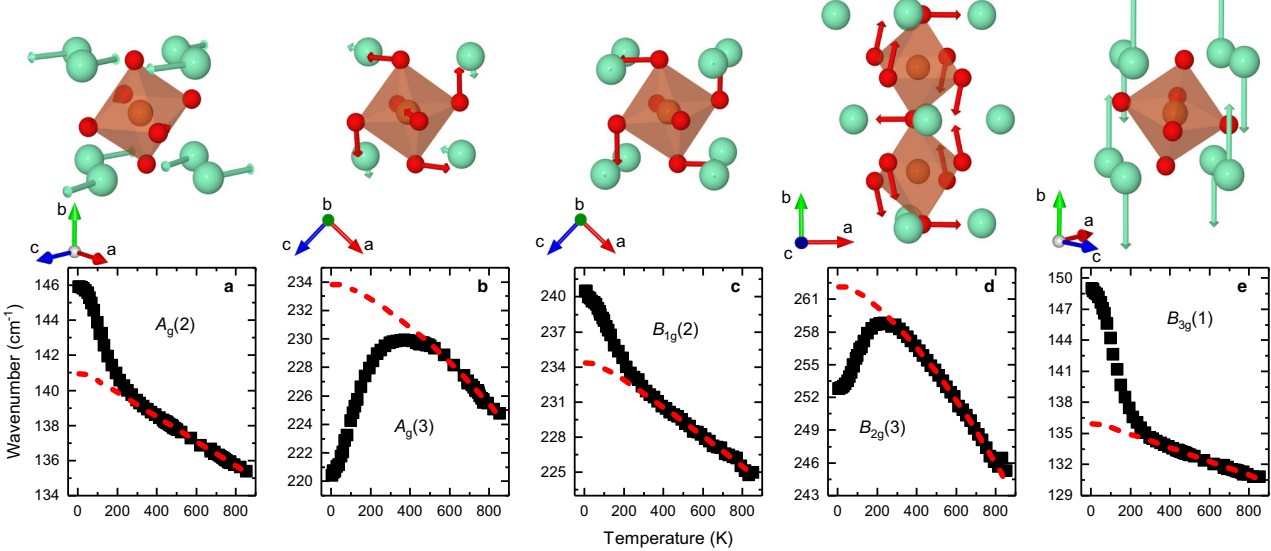

**Fig. 3 Anomalous deviation of lattice vibrations.** Temperature dependent frequencies of five different vibrations that show an anomalous evolution resulting from the induced $Sm^{3+}$ spin ordering. The displacement patterns[56] involve pure samarium displacements **a**, **e**, in-phase **b**, **c** and out-of-phase **d** rotations of the octahedra. The dashed red lines indicate the evolution expected from an undisturbed anharmonic frequency evolution. The Raman spectra that provide the basis for these data were measured in the following light polarization settings: $A_{g}$: $x(zz)\overline{x}$, $B_{1g}$: $z(xy)\overline{z}$, $B_{2g}$: $y(xz)\overline{y}$, $B_{3g}$: $x(yz)\overline{x}$.

magnetic order parameter remains small throughout the full temperature range, including through the Néel point and the spin-reorientation transition.

**Induced $Sm^{3+}$ ordering ($T < 300$ K).** Having observed emergent magnetoelastic properties as the samarium acts on the magnetic order of iron, we now turn to the reverse case with iron acting on samarium. Upon cooling below the spin reorientation, the iron magnetism induces a non-spontaneous ordering of the $Sm^{3+}$ spins. Our Raman spectra reveal two types of spectral anomalies in this regime. First, the frequency evolution of several vibrational bands dramatically deviates with temperature from a typical anharmonic behavior (Fig. 3). Second, we observe the emergence of new Raman features (Fig. 4). In comparison with the Raman results, RUS is virtually featureless with only subtle anomalies at 250 K that could be consistent with magnetoelastic coupling (see Supplementary Note 8).

The observed deviations of the Raman frequencies from a typical thermal behavior are unprecedentedly strong in perovskites. While spin–phonon coupling commonly leads to Raman shifts of a few wavenumbers at most[29,30,36], we observe deviations of over $10 \, cm^{-1}$ and up to 8%. To understand the underlying mechanism of the deviations, we take a closer look at the vibrational pattern of the affected Raman modes. The low frequency $B_{3g}(1)$ and $A_{g}(2)$ modes are pure samarium vibration modes along the $y$- and $z$-axis, respectively, while $B_{1g}(2)$, $A_{g}(3)$ and $B_{2g}(3)$ are rotation modes of the octahedra, which include samarium displacements. The anomalies of the pure samarium vibrations ($B_{3g}(1)$ and $A_{g}(2)$) are a clear sign of a modification of the $Sm^{3+}$ sublattice emerging from the non-spontaneous alignment of the samarium spins. Changes of the tilt vibrations, $B_{1g}(2)$, $A_{g}(3)$ and $B_{2g}(3)$, are, at first sight, less intuitive to relate to the $Sm^{3+}$ spin ordering. However, in the same way as the rotations of the octahedra affect the interaction of neighboring iron spins, $FeO_6$ rotations alter the Fe-O-Sm coupling path. The orientation of the samarium-spin sublattice is steered by the trilinear coupling of $Fe^{3+}$ spins, $Sm^{3+}$ spins and the octahedral tilt system, as proposed theoretically by Zhao and co-workers[14].

Hence, the effective field acting on the $Sm^{3+}$ spins is of magneto-structural origin and links the samarium magnetism to the rotations of the octahedra. The FWHM that are commonly more susceptible to coupling phenomena show anomalies for all phonon modes below $350 \, cm^{-1}$, i.e. modes that include $Sm^{3+}$ motions. Only motions in $x$-direction are not affected, which epitomises the naturally anisotropic character of the $Sm^{3+}$-$Fe^{3+}$ spin interaction. For a further discussion see Supplematary Note 2. Overall, the interaction between $Sm^{3+}$ and $Fe^{3+}$ gives rise to the emergence of spin–phonon coupling, which we probe as a strong anomaly of samarium and octahedron vibrations.

New Raman features are highlighted in Fig. 4 as red-shaded areas. These features are not observed at ambient conditions but emerge gradually with decreasing temperature at 112 and $224 \, cm^{-1}$ in $z(xy)\overline{z}$ as well as 130, and $287 \, cm^{-1}$ in $y(xz)\overline{y}$ configuration, using Porto's notation[37]. To better illustrate the emergence, we show in Fig. 4e how the features at 130 and $287 \, cm^{-1}$ gain in intensity with decreasing temperature. (For further emergent feature see Supplementary Note 3).

To understand the consequences that come with the emergence of new Raman features, we need to identify their nature. Firstly, the new bands cannot be previously masked vibration modes, since all vibrational bands of *Pnma* symmetry in $SmFeO_3$ have been identified[24]. We need to consider the following typical origins for the new features:

(i) $Sm^{3+}$ crystal-field excitations: Because of the non-centrosymmetric position of $Sm^{3+}$ ions, the ground-state energy of samarium is split. The interaction between rare-earth and iron can impact these low energy levels as earlier investigations of $ReFeO_3$ show[38–43]. However, such bands have only been observed by submillimeter spectroscopy and low frequency Raman modes are limited to the well-known magnon excitations of the $Fe^{3+}$-spin[7]. Overall, electronic $Sm^{3+}$ excitations are expected at frequencies below $100 \, cm^{-1}$, much lower than the position of our new Raman active features. Therefore, we exclude low energy $Sm^{3+}$-transitions as origin of the new features.

(ii) Two-magnon bands result from scattering of two magnetic excitations not only at the $\Gamma$-point but also at the zone boundary. Therefore, they can be found at higher frequencies. Resulting

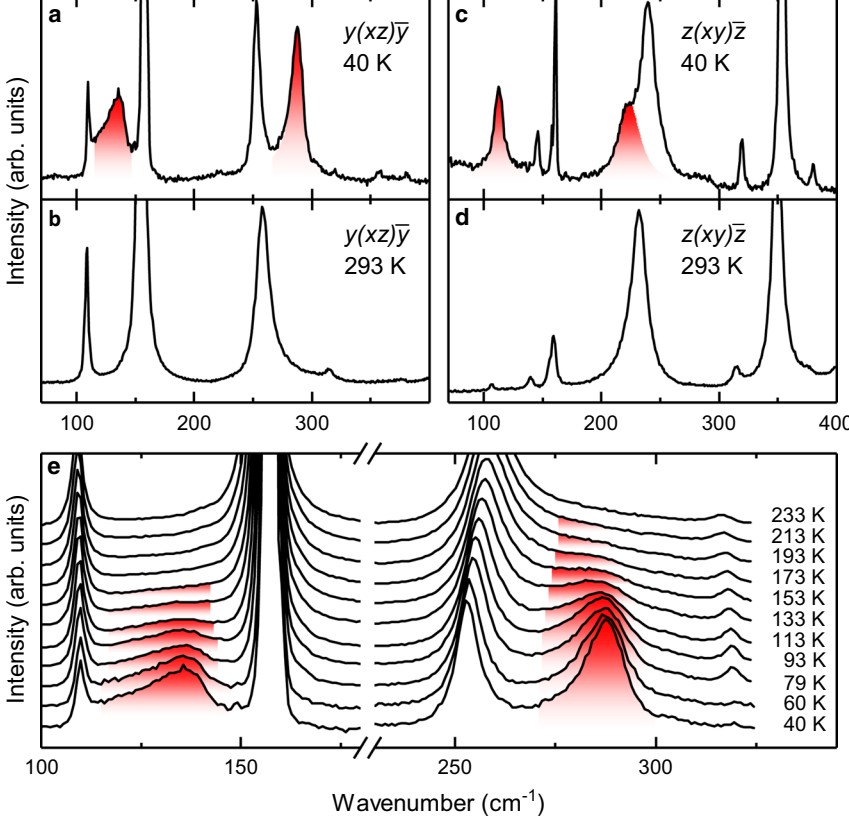

**Fig. 4 Emerging of new phonon modes.** Raman spectra for $y(xz)\bar{y}$, **a** and **b**, and $z(xy)\bar{z}$, **c** and **d**, configurations at 40 and 293 K, respectively. **e** Temperature dependent Raman spectra in $y(xz)\bar{y}$ configuration. The colored areas indicate new Raman-active bands.

from a second order process, two-magnon bands are known to show broader and more asymmetric shapes than first-order processes, see for example in TbMnO$_3$ upon the transition into the incommensurate multiferroic phase[44]. However, since the single-magnon bands do not show an anomalous behavior below room temperatures, a magnonic nature of the new bands is unlikely (see Supplementary Note 7).

(iii) New phonons: The shape of the emerging bands at 112, 224 and 287 cm$^{-1}$ strongly resemble the neighboring phonon modes. In addition, these features emerge simultaneously with the anomalies of the vibrational modes. Excluding the previous options, we therefore assign these bands as new phonon bands.

To put these findings into context, we compare them with well-known systems that experience a magnetically induced phase transition. In collinearly antiferromagnetic $R$Mn$_2$O$_5$ the symmetry breaking and induced ferroelectricity are barely detectable in the phonon spectrum (deviations < 0.5 cm$^{-1}$) and no new Raman bands are observed[45–48]. Likewise, phonon deviations in spin-spiral systems are below the resolution limit, e.g. in MnWO$_4$[49]. In TbMnO$_3$, the symmetry breaking can only be hypothesized from shifts of the vibration frequencies smaller than 1.5 cm$^{-1}$ and new extremely broad features are assigned to two-magnon excitations[50]. Overall, anomalies in SmFeO$_3$ driven by the interplay of iron and samarium exceed these examples by one order of magnitude. Furthermore, it is instructive to compare SmFeO$_3$ to its ferroelectric sibling GdFeO$_3$. Unlike SmFeO$_3$ where the Sm$^{3+}$ sublattice order is induced, the ferroelectric phase transition in GdFeO$_3$ results from the independent and spontaneous ordering of both, the Gd$^{3+}$ and the Fe$^{3+}$ sublattices. The changes to the Raman spectra, however, show identical characteristics. With the ordering of the magnetic Gd$^{3+}$ spins, Gd$^{3+}$ and octahedral tilt vibrations show the same anomalous

deviations as in SmFeO$_3$ and the emergence of new bands[51]. These similarities reinforce the assignment of the new features as phonons. Yet, the physical origins of the phenomena in GdFeO$_3$ and SmFeO$_3$ are strikingly different resulting from classical spontaneous Gd$^{3+}$-Gd$^{3+}$ ordering, as opposed to non-spontaneous, but iron-induced Sm$^{3+}$ ordering. Therefore, in GdFeO$_3$ the anomalies are limited to the ferroelectric phase below 2.5 K[9], while in SmFeO$_3$ they occur at two orders of magnitude higher temperatures.

Furthermore, the emergence of new phonon bands itself, in direct analogue to GdFeO$_3$, provides evidence for a change of phase in SmFeO$_3$ through a symmetry lowering. Raman spectroscopy does not allow for an identification of the symmetry. From the evolution of the phonon modes and the lattice constants (see Supplementary Note 9), we can estimate, however, that the material remains orthorhombic with the possible point groups *222* or *mm2*. This consequence of the Fe$^{3+}$-Sm$^{3+}$-interplay is astonishing and goes beyond a rise of strong spin–phonon coupling.

In conclusion, we scrutinized the spin–phonon coupling in SmFeO$_3$, a model material for the interaction of two magnetic ions, throughout all magnetic phases. Any coupling between the magnetic order parameters of iron alone and strain is weak, such that there are no obvious anomalies in the evolution of elastic moduli through the Néel critical point, on either side of the spin reorientation transition, or associated with the magnetic cluster glass formation. This is a reflection, primarily, of the fact that Fe$^{3+}$ is not Jahn-Teller active. On the other hand, there is a significant anelastic effect through the temperature interval of the spin reorientation transition, which is ascribed to relaxational magnetoelastic effects of locally strained regions that might possibly be monoclinic.

Once the $Sm^{3+}$ and $Fe^{3+}$ spins start interacting, however, a strong spin–phonon coupling emerges. This coupling manifests in the anomalous evolution of vibrational bands and in the emergence of new Raman active modes. It is activated by the non-spontaneous, though intrinsic ordering of the $Sm^{3+}$ spins in the exchange field of the magnetic $Fe^{3+}$-sublattice. We observe strong indications—identical to the ferroelectric phase transition in $GdFeO_3$—that this non-spontaneous ordering induces a phase change in $SmFeO_3$. In addition, our findings support the theoretical prediction of the trilinear coupling between $Fe^{3+}$ and $Sm^{3+}$ spins and the $FeO_6$ tilt vibrations[14]. While this seminal theoretical work focuses on the influence of the tilts on the magnetism, we demonstrate here the reverse-effect of the magnetism on the structural vibrations.

We have shown how the non-linear interplay of two magnetic orders can trigger significant variations of lattice motions. We expect that the presented effects are not limited to $SmFeO_3$, but likely exist in a vast variety of systems like rare-earth manganites, ferrites or chromites, where magnetic transition-metal sublattices impose a magnetic ordering on rare-earth sublattices. We have shown that such non-linear coupling of magnetic orders can give rise to enhancement phenomena and even new phases exceeding by far the sum of the initial properties. We expect that such phenomena are not limited to magnetic orders but may play a role for a large number of interacting orders. Importantly, these effects emerge just below room temperature and are not limited to cryogenic temperatures which makes them attractive for potential applications. This work motivates the specific search for hidden non-linear material responses, in experiment and theory, to achieve a conclusive picture of the microscopic interaction mechanisms at play and their potential exploration in technology applications.

## Methods

**Sample preparation**. $SmFeO_3$ single crystal samples were grown in a four-mirror optical-floating-zone furnace (FZ-T-10000-H-VI-P-SH, Crystal Systems Corp.) as described elsewhere[19]. Crystals of all three orthorhombic orientations were prepared, lapped to a thickness of 80 μm and polished optically flat.

**Raman spectroscopy**. Raman spectroscopy measurements were performed with an inVia Renishaw Reflex Raman Microscope in micro-Raman mode with a 633-nm He-Ne laser. We avoided sample heating by limiting the laser power. Frequencies and FWHM of the phonon modes were obtained by fitting the Raman spectra with Lorentzian functions. During the Raman scattering experiments, the temperature of the crystals was controlled using a Linkam THMS600 stage and an Oxford Instruments Microstat for cryogenic temperatures.

**Resonant ultrasound spectroscopy**. The technique of Resonant Ultrasound Spectroscopy (RUS) involves the measurement of acoustic resonances of mm-sized samples between two piezoelectric transducers and has been described in detail by Migliori and Sarrao[26]. The first transducer excites mechanical vibrations, typically in the frequency range 0.01–1 MHz, and the second detects resonances at frequencies which depend on the size, shape and density of the sample and on the values of its elastic moduli.

Individual peaks in the primary spectra are fitted to determine their frequency, $f$, and width at half maximum height $\Delta f$. Each resonance is typically dominated by shearing motions and the square of the resonance frequencies scales with different combinations of the (predominantly shear) elastic moduli. Acoustic loss is expressed in terms of the inverse mechanical quality factor, $Q^{-1}$, which is taken to be $\Delta f/f$.

Two different instruments were used for measurements above and below room temperature. In the low temperature instrument, the sample sits directly between the transducers and the holder is lowered into a helium flow cryostat[52]. A few mbars of helium are added to the sample chamber to assist thermal equilibration between the sample and the cryostat. In the high temperature instrument, the sample sits between the tips of alumina buffer rods which are inserted into a horizontal resistance furnace with the transducers attached to the ends of the rods, outside the furnace[53].

An irregular fragment with dimension ~1 mm³ and mass 0.0158 g was selected for study on the basis that it did not show any externally visible cracks. Spectra containing 65,000 data points were collected in automated cooling/heating sequences using a settle time of 20 min at each set point to allow for thermal equilibration. Liquid nitrogen was used for cooling down to ~110 K. For the high temperature measurements, the sample was held in an argon atmosphere. Selected peaks in the primary spectra were fitted with an asymmetric Lorentzian function to extract values of $f$ and $\Delta f$ as a function of temperature, using the software package Igor (Wavemetrics).

## Data availability

The Raman spectroscopy and Resonant Ultrasound Spectroscopy data generated in this study have been deposited in the Research Collection database of the ETH Zurich under accession code https://doi.org/10.3929/ethz-b-000512405[54].

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

## Acknowledgements

The authors thank Christian Tzschaschel, Steven Huband and Inmaculada Peral Alonso for fruitful discussions. MCW and MF are grateful for financial support from the SNSF (Grant No. 200021_178825), European Research Council (Advanced Grant 694955-INSEETO) and the SNSF Spark funding CRSK-2_196061. MCW also for the support from the SNSF Spark funding CRSK-2_196061. MCW, MG, CT and JK acknowledge financial support from the Fond National de Recherche Luxembourg through a PEARL grant (FNR/P12/4853155/Kreisel), BD through an INTER mobility grant (INTER/Mobility/19/13992074) and the project EXPAND (ANR-17-CE24-0032) and AS and YK through the SNF Ambizione PZ00P2_180035 grant. RUS facilities were established and supported through grants from the Natural Environment Research Council (NE/B505738/1, NE/F017081/1) and the Engineering and Physical Sciences Research Council (EP/I036079/1, EP/P024904/1) awarded to MAC. SC and WR are grateful for financial support from the National Natural Science Foundation of China (NSFC, Nos. 12074242, 51911530124), and the Science and Technology Commission of Shanghai Municipality (No.21JC1402600).

## Author contributions

M.C.W. coordinated the project and performed and analyzed the Raman spectroscopy measurements supported by CT and guided by M.G.; RUS measurements were performed and analyzed by M.A.C., D.M.E., M.C.W. and M.G.; A.S. performed and analyzed the X-ray diffraction measurements together with Y.K.; B.D. and J.K. initiated the project. X.M., W.R. and S.C. grew, cut, and characterized the high quality $SmFeO_3$ single crystal samples used in the experiments. J.K. and M.F. supervised the work. All authors contributed to the discussion and writing of the manuscript.

## Competing interests

The authors declare no competing interests.
