## [Peer Review File · Nature Communications]

REVIEWER COMMENTS

Reviewer #1 (Remarks to the Author):

The paper presents an interesting study of the interplay of two magnetic ordering in the case of SmFeO_3 , which induces a strong spin-phonon coupling. The induced Sm^{3+} ordering (below 300K) is well explained. I have some questions/comments that should be clarified concerning the Raman results in the 300-800 K temperature range, and the appearance of the new modes at lower temperature:

1. The FWHM of Ag(3) (Figure S1) has nearly the same values at 300 K and 800 K (which is a rather unusual behaviour). It will be very interesting to show some Raman spectra measured at different temperatures in this temperature range, to get an idea about the temperature evolution of the relative intensities and FWHM of the other Ag-modes.
2. On the other hand, below TSR the FWHM of the Ag(4) mode decreases with decreasing temperature (figure S1 b), while for the Ag(3) the FWHM remains constant (figure S1 a). Why is the behaviour for the two modes different below TSR? Considering the connection between FWHM and the phonon lifetime across the spin reorientation mentioned by the authors in the paper (line 143-144), why the modes should behave differently at lower temperatures?
3. Are the results shown in the figure 3 resulting from polarised Raman spectra? If yes, from which polarisation configuration?
4. The authors proposed that the new phonon bands “provide evidence for a change of phase in SmFeO_3 ”. (lines 236-239) Which is the new phase? Are there any evidence for structural transitions for SmFeO_3 in this temperature range (temperature dependent XRD or neutron scattering)?
5. Are the new emerging modes (shown in Figure 4) appearing only in cross polarisation configuration, or also in parallel configuration? If the authors assign these band as new phonon bands (lines 217-218), do they have any polarisation dependence?
6. Technical comment:

Caption Figure 2 b,c (lines 129-130): "Evolution of the frequency (black squares) and FWHM (blue circles) of Ag(3) and Ag(4) Raman modes from 300 to 850 K, across the spin reorientation and Néel temperatures."

In the figure 2b,c there are no blue circles, maybe the authors refer to the Figure S1?

Reviewer #2 (Remarks to the Author):

This manuscript reports new phonon modes induced by Fe³⁺ - Sm³⁺ spin-spin interactions in SmFeO₃. This material has received a considerable attention over a decade owing to its interesting coupling behavior between different order parameters. The present work shows that the strong spin-phonon coupling emerges once the Sm³⁺ and Fe³⁺ spins start interacting. This is a well-written manuscript. In addition, the studies are well-done and provide useful insight to those working in the field of superconductivity and multiferroics with oxides. However, the spin-phonon coupling phenomenon in GdFeO₃ has already been reported as the authors mentioned even though the exact mechanism is a bit different (there is no Fe spin-induced Gd spin ordering). As a result, I don't find this manuscript sufficiently novel or significant to appeal to a broader audience or warrant publication in Nature Communications. My recommendation is for submission to a more specialized journal such as physical review journals with the current form.

Reviewer #3 (Remarks to the Author):

The authors have demonstrated the 3d-4f dynamics of iron and samarium in samarium orthoferrite single crystal. They evidenced it by observing large shift in phonon frequencies as well as noticing new phonons. This strong spin phonon coupling is attributed to ordering of samarium spins, which is induced by iron spins.

The anomalous deviation of lattice vibrations and emergence of new Raman bands at considerably higher temperatures (~ 200 K) is very intriguing and the appearance of bands is significant. This finding is new and indicative of new possibilities in such systems.

Interestingly, it is demonstrated that lattice vibrations are induced due to magnetic ordering of Sm and Fe ions. Consequently, this work establishes a rare occurrence of strong spin phonon coupling in the samarium orthoferrite at a temperature which is significantly higher than the cryogenic temperature and not reported earlier.

The manuscript is written in an impressive manner with very logical subheadings. The explanation, analyses and justification of data are conveyed in very structured manner to the reader. Figures are appropriate and indicating the significant findings. The conclusion stimulates the researchers working in similar materials. The work will be useful to understand and explore the dynamics and interplay between rare earth ion and transition metal ion leading to spin phonon coupling and its consequences in other potential material.

I recommend the manuscript for publication and congratulate the authors for this unusual and extraordinary finding.

Response to the Reviewers' remarks:

We thank the Reviewers for their careful analysis and time invested into reviewing our manuscript, for their benevolent reception of our work and their constructive comments. In the following, we address point-by-point the input and suggestions for improvement. The reviewers' scientifically interesting and important questions have led to substantial additions mainly to the supplementary information and the new X-ray diffraction experiments at the synchrotron. These new measurements caused the delay in resubmission and led to two additional authors, namely Arkadiy Simonov and Yevheniia Kholina. We are convinced that the changes and additions to the manuscript improved of the manuscript substantially.

Changes to the manuscript are marked in blue for easy tracking.

Reviewer #1 (Remarks to the Author):

Comment 1.1: The paper presents an interesting study of the interplay of two magnetic ordering in the case of SmFeO₃, which induces a strong spin-phonon coupling. The induced Sm³⁺ ordering (below 300K) is well explained. I have some questions/comments that should be clarified concerning the Raman results in the 300-800 K temperature range, and the appearance of the new modes at lower temperature:

Response 1.1: We thank the reviewer for the careful analysis of the work and appreciate the time invested to form scientifically interesting and important questions. The questions have led to substantial additions to the manuscript, which certainly improve the quality of the manuscript for all readers.

Comment 1.2: The FWHM of A_g(3) (Figure S1) has nearly the same values at 300 K and 800 K (which is a rather unusual behaviour). It will be very interesting to show some Raman spectra measured at different temperatures in this temperature range, to get an idea about the temperature evolution of the relative intensities and FWHM of the other A_g-modes.

Response 1.2: This is an interesting point. To answer it in the best possible way, we have included additional data sets with even higher signal to noise ratio to optimize our resolution. The extensive additional data - Raman spectra, phonon frequencies and FWHM - are given in Supplementary Information 1 and 2.

As we see from the additional data, the anomalies in the evolution of the FWHM mentioned by the Reviewer occur for modes that are impacted by the emergent interaction of Sm^{3+} and Fe^{3+} . Hence, the anomalies are a direct consequence of coupling of the phonons with the two magnetic sublattices.

Comment 1.3: On the other hand, below TSR the FWHM of the $A_g(4)$ mode decreases with decreasing temperature (figure S1 b), while for the $A_g(3)$ the FWHM remains constant (figure S1 a). Why is the behaviour for the two modes different below TSR? Considering the connection between FWHM and the phonon lifetime across the spin reorientation mentioned by the authors in the paper (line 143-144), why should the modes behave differently at lower temperatures?

Response 1.3: The differences visible in the $A_g(3)$ and $A_g(4)$ modes are associated with two different scenarios. i) If the phonon modes are strongly impacted by the Sm^{3+} - Fe^{3+} interaction i.e., the phonon frequencies show clear deviations, the FWHM of the phonons show a reduced slope (or even an increase in value) prior to a rapid decline. ii) If the phonon frequencies are less affected by the Sm^{3+} - Fe^{3+} interaction, the decrease of the FWHM is more gradual. The $A_g(3)$ and the $A_g(4)$ modes are members of the former and latter scenarios, respectively.

Changes to the manuscript 1.3: We added the temperature evolutions of the FWHM together with a more detailed discussion to the Supplementary Information 1 and 2 and to the main text in lines 180 and following and in lines 130-131.

Comment 1.4: Are the results shown in the figure 3 resulting from polarised Raman spectra? If yes, from which polarisation configuration?

Response 1.4: Yes, all measurements were performed for specific analyzer and polarizer settings and the corresponding spectra are given in Supplementary Information 1.

Changes to the manuscript 1.4: We have now noted all the settings in the caption of figure 3, lines 191 and 192.

Comment 1.5: The authors proposed that the new phonon bands “provide evidence for a change of phase in SmFeO_3 ”. (lines 236-239) Which is the new phase? Are there any evidence for structural transitions for SmFeO_3 in this temperature range (temperature dependent XRD or neutron scattering)?

Response 1.5: This is an excellent point. From the known orderings of the magnetic sublattices (*Nat. Commun.* **8**, 14025 (2017)), we can assume that SmFeO₃ remains orthorhombic with the point groups 222 or *mm2*. Yet, the Raman-active phonons show the same anisotropic dependencies for the space groups *mmm*, 222 or *mm2*. Therefore, anisotropy measurements do not allow to distinguish these of the phases.

Following the reviewer's suggestion, we conducted both XRD and neutron scattering measurements. The neutron scattering experiments at the ILL, to identify the magnetic order of samarium and with this the overall symmetry, were hampered by the strong neutron absorption of naturally occurring Sm¹⁴⁹. SmFeO₃ with specific samarium isotope concentrations to circumvent the neutron absorption was not available.

To identify structural changes, we performed synchrotron single crystal X-ray diffraction measurements at the ESRF. These new experiments led to two additional authors, namely Arkadiy Simonov and Yevheniia Kholina. We show the temperature evolution of the lattice parameters in Supplementary Information 9. We find a significant anomalous deviation of the *c*-axis lattice parameter and of the spontaneous strain that coincide with the phonon anomalies. Hence, the anomalies are equally mirrored in the structural data. This is remarkable since X-ray diffraction is not a probe that is highly sensitive to magnetically induced symmetry changes. Yet, a slight twinning of the crystals prevented a full refinement of the structure and the identification of the new phase. Therefore, we avoid speculation on the nature of the emergent phase - as it goes beyond the scope of this work.

Change to the manuscript 1.5: For clarification we added a comment to lines 244 and following. The X-ray data and their discussion are given in the Supplementary Information 9.

Comment 1.6: Are the new emerging modes (shown in Figure 4) appearing only in cross polarisation configuration, or also in parallel configuration? If the authors assign these band as new phonon bands (lines 217-218), do they have any polarisation dependence?

Response 1.6: We observed emerging modes for crossed as well as parallel light polarization configurations (now added to the Supplementary Information 3). The new phonon bands show a clear polarization dependence. Just like the original phonons, the new bands are only accessible under specific light polarization configurations as given in the corresponding figures.

Change to the manuscript 1.6: We added further data with a new emerging phonon for a parallel configuration of the incident and probed light polarizations (Supplementary Information 3) and make the corresponding references in the main text (line 201).

Comment 1.7: Technical comment: Caption Figure 2 b,c (lines 129-130): “Evolution of the frequency (black squares) and FWHM (blue circles) of Ag(3) and Ag(4) Raman modes from 300 to 850 K, across the spin reorientation and Néel temperatures.” In the figure 2b,c there are no blue circles, maybe the authors refer to the Figure S1?

Response 1.7: Thank you, we fixed this issue.

Change to the manuscript 1.7: The captions to Figure 2 and Figure S2 have been updated.

Reviewer #2 (Remarks to the Author):

Comment 2.1: This manuscript reports new phonon modes induced by Fe³⁺ - Sm³⁺ spin-spin interactions in SmFeO₃. This material has received considerable attention over a decade owing to its interesting coupling behavior between different order parameters. The present work shows that the strong spin-phonon coupling emerges once the Sm³⁺ and Fe³⁺ spins start interacting. This is a well-written manuscript. In addition, the studies are well-done and provide useful insight to those working in the field of superconductivity and multiferroics with oxides.

Response 2.1: We thank the reviewer for the overall positive assessment of our work.

Comment 2.2: However, the spin-phonon coupling phenomenon in GdFeO₃ has already been reported as the authors mentioned even though the exact mechanism is a bit different (there is no Fe spin-induced Gd spin ordering). As a result, I don't find this manuscript sufficiently novel or significant to appeal to a broader audience or warrant publication in Nature Communications. My recommendation is for submission to a more specialized journal such as physical review journals with the current form.

Response 2.2: While the results of the couplings are similar, the respective mechanisms are completely different. We believe that this difference is one of the key messages of our paper and we realize that we have to make this clearer.

In GdFeO₃, at 2.5 K, the spin-phonon coupling sets in due to the Gd-Gd spin ordering, which breaks the inversion symmetry and induces a ferroelectric order through exchange-striction. *This order is spontaneous*. Hence, spin-phonon coupling and the emergence of new phonon bands are therefore expected.

In SmFeO₃, at ca. 200-250 K, the origin of the spin-phonon coupling, as the reviewer writes, results from the samarium spin order induced by the iron magnetism. *This order is non-spontaneous*. This mechanism leads to spin-phonon-coupling effects that are one order of magnitude stronger and occur at two orders of magnitude higher temperatures than in the spontaneously ordered GdFeO₃.

As such, the origin of the coupling, its strength, and the 'operating' temperatures are completely different. Indeed, so far rare-earth ordering phenomena (in rare-earth transition metal oxides) are restricted to effects below 10 K and are, therefore, of purely academic interest. In contrast, the mechanism observed in SmFeO₃ shows that rare-earth ordering phenomena can occur at temperatures that can conveniently

be reached e.g., with Peltier elements or CO₂. With the possibility to scale rare-earth ordering effects to higher temperatures, these compounds become interesting for applications. Therefore, the phenomena emerging from non-spontaneous ordering are also of fundamental significance. From the authors' perspective, this is highly exciting as it means that spin-phonon-coupling might be only one of many phenomena – still to be discovered – that occur at “high”-temperatures where an order/property is “self-induced”.

Change to the manuscript 2.2: We added further clarification to the text in lines 238 and following.

Reviewer #3 (Remarks to the Author):

Comment 3.1: The authors have demonstrated the 3d-4f dynamics of iron and samarium in samarium orthoferrite single crystal. They evidenced it by observing large shift in phonon frequencies as well as noticing new phonons. This strong spin phonon coupling is attributed to ordering of samarium spins, which is induced by iron spins.

The anomalous deviation of lattice vibrations and emergence of new Raman bands at considerably higher temperatures (~ 200 K) is very intriguing and the appearance of bands is significant. This finding is new and indicative of new possibilities in such systems.

Interestingly, it is demonstrated that lattice vibrations are induced due to magnetic ordering of Sm and Fe ions. Consequently, this work establishes a rare occurrence of strong spin phonon coupling in the samarium orthoferrite at a temperature which is significantly higher than the cryogenic temperature and not reported earlier.

The manuscript is written in an impressive manner with very logical subheadings. The explanation, analyses and justification of data are conveyed in very structured manner to the reader. Figures are appropriate and indicating the significant findings. The conclusion stimulates the researchers working in similar materials. The work will be useful to understand and explore the dynamics and interplay between rare earth ion and transition metal ion leading to spin phonon coupling and its consequences in other potential material.

I recommend the manuscript for publication and congratulate the authors for this unusual and extraordinary finding.

Response 3.1: We thank the reviewer for their careful reading of our manuscript and the corresponding clear and positive analysis.

REVIEWERS' COMMENTS

Reviewer #1 (Remarks to the Author):

All the concerns/comments raised in my previous review have been addressed in the revision.

The new results, the changes and additions to the manuscript (Supplementary Information) brought a clear improvement clarifying some points/concerns. I recommend publication of the manuscript.

Reviewer #2 (Remarks to the Author):

I'm OK with the response letter and revised manuscript.